# A Novel Beta-Glucosidase Gene for Plant Type Was Identified by Genome-Wide Association Study and Gene Co-Expression Analysis in Widespread Bermudagrass

**DOI:** 10.3390/ijms231911432

**Published:** 2022-09-28

**Authors:** Lu Gan, Minghui Chen, Jingxue Zhang, Jibiao Fan, Xuebing Yan

**Affiliations:** 1College of Animal Science and Technology, Yangzhou University, Yangzhou 225009, China; 2College of Economics and Management, Nanjing Forestry University, Nanjing 210037, China

**Keywords:** bermudagrass, plant type, GWAS, IAA, BGLU

## Abstract

Bermudagrass (*Cynodon* spp.) is one of the most widely distributed warm-season grasses globally. The growth habits and plant type of bermudagrass are strongly associated with the applied purpose of the landscape, livestock, and eco-remediation. Therefore, persistent efforts are made to investigate the genetic basis of plant type and growth habits of bermudagrass. Here, we dissect the genetic diversity of 91 wild bermudagrass resources by genome-wide association studies (GWAS) combined with weighted gene co-expression analysis (WGCNA). This work is based on the RNA-seq data and the genome of African bermudagrass (*Cynodon transvaalensis* Burtt Davy). Sixteen reliable single-nucleotide polymorphisms (SNPs) in transcribed regions were identified to be associated with the plant height and IAA content in diverse bermudagrass by GWAS. The integration of the results from WGCNA indicates that *beta-glucosidase 31* (*CdBGLU31*) is a candidate gene underlying a G/A SNP signal. Furthermore, both qRT-PCR and correlation coefficient analyses indicate that *CdBGLU31* might play a comprehensive role in plant height and IAA biosynthesis and signal. In addition, we observe lower plant height in Arabidopsis *bglu11* mutants (homologs of *CdBGLU31*). It uncovers the breeding selection history of different plant types from diverse bermudagrass and provides new insights into the molecular function of *CdBGLU31* both in plant types and in IAA biosynthetic pathways.

## 1. Introduction

Grasses are major components of many terrestrial ecosystems and are widely distributed in grasslands, urban landscapes, and other ecological spaces. Like many other cereals, bermudagrass (*Cynodon* spp.) is an essential warm-season grass of the *Poaceae* family. Due to its diverse growth habits and strong ability to withstand stress, bermudagrass is widely used in landscapes as lawn grass, for grazing livestock as a forage grass, and on barren land as ecological grass. It is indicated that bermudagrass is likely “multipurpose” in the green ecosystem. Moreover, bermudagrass are well known for their tremendous diversity in plant forms, application value, and extraordinary diversity in growth habits and chromosome numbers of different genotypes [1,2,3].

Since the 1960s, breeders have attempted to distinguish genetic traits and select superior genes with greater yields and better quality in breeding programs exploring the tremendous genetic diversity of the bermudagrass collection [4,5,6]. For further improvements, breeders require more parent material with morphological and genetic diversity to maximize genetic gains for desirable traits. Unlike traditional analysis of genetic diversity (e.g., DNA-amplification fingerprinting, amplified fragment length polymorphism, and simple sequence repeat), recent genome sequencing efforts of African bermudagrass support a substantial genetic contribution to the study of the genetic diversity and key genes for desirable traits based upon the GWAS and transcription-wide association study (TWAS) [7]. Transcriptome analysis has been extensively used to identify differentially regulated genes and characterize transcriptional changes. The transcriptomics data can also be mined for genetic variation, including single-nucleotide polymorphisms (SNPs) [8,9]. These polymorphic markers of transcribed regions can perform association analyses (e.g., GWAS and TWAS) to identify marker–trait associations with plant phenotype.

As an important warm-season forage and turfgrass, bermudagrass is more variable in genome size and ploidy level, so the genetic and morphological diversity of abundant wild germplasm resources provides different varieties for bermudagrass breeding [10,11]. According to the development levels of shoots and stolons, wild bermudagrass are commonly divided into forage-types and turf-types based on different plant types [12,13,14,15]. To produce more forage and digestible nutrients, forage-type accessions of bermudagrass may represent more leaf content with high shoots and less stems with weak stolons [6,16]. To fulfill the requirement of turf and soil stabilization, turf-type cultivars of bermudagrass usually have prostrate growth habits with stolons and lateral-spreading shoots [17]. So, elucidating the regulated mechanism of the growth and development of stolons and shoots in bermudagrass not only explains the features of different plant architecture but also provides helpful information for breeding new cultivars with different applications [15]. In order for stolons, shoots, and leaves to grow and develop, phytohormones, growth factors, and steroids all work synergistically with each other. Balatti and Willemoes reported the relationship between ethylene and the gravity-induced upward bending of bermudagrass stolons [18]. According to Yun et al., the Aux/IAA auxin signal factor gene may play a role in rhizome development [19]. However, genome-wide associate studies focusing on the mechanism of architectural regulation in bermudagrass have not been reported.

In this study, we obtained the transcriptomes of 91 wild individuals of bermudagrass with different architecture. By employing high-density SNP genotyping and RNAseq data, our study aimed to explore associated SNP markers with plant height and IAA content via GWAS and identify candidate genes at GWAS-identified loci via differential expression analyses.

## 2. Results

### 2.1. Delineating Relationship among Plant Height and IAA Content with the Latitudinal Gradient in Diversity Bermudagrass

We selected a diverse panel of 91 wild bermudagrass germplasms covering subtropical and mesothermal zones ranging from 22°35′40″ to 36°18′40″ in China’s 16 geographical locations (Appendix A). We observed high diversity in plant architecture and phenotypic data ranging from low to high latitude (Figure 1a). The study of germplasm resources of bermudagrass from the subtropical zone found that plants’ natural height is negatively correlated with latitude (*r*^2^ = 0.89), among which the L10 samples performed prostrate growth with the shortest natural height (Figure 1a,b). In contrast, the samples from the mesothermal region showed vertical growth, and there was a significant positive correlation between plant height and latitude (*r*^2^ = 0.957). Additionally, the content of indoleacetic acid (IAA), which impacts plant height and architecture, was related to the latitude in samples from the subtropical zone (*r*^2^ = 0.792, Figure 1c).

### 2.2. Application of GWAS to Identify the Genetic Regions That Influence Plant Height and IAA Content in Diverse Bermudagrass

In this study, we combined MLM, GLM, and FarmCPU models to determine the association between two plant traits and 4,751,646 SNPs from RNAseq data mapped to the African bermudagrass reference. We used log10(*p*) > 5 as the genome-wide significance threshold for the association between plant height and SNPs after a Bonferroni correction (Figure 2a). Subsequently, we found a synonymous G/A SNP in 36,689,561 bp (encoding a beta-glucosidase) on chromosome 9, where an altered allele (A) may be related to plant height (Table 1). Furthermore, a hotspot region on chromosome 3 was predicted to be located in the *LG03.3214* gene encoding the Pom33 family.

We also performed a GWAS at the threshold of log10(*p*) > 8 on the IAA content with SNPs as an independent to identify SNPs that might influence the auxin biosynthesis and plant height (Figure 2b). We found highly significant association signals on chromosome 4 (*evm.model.LG04.1964* encoding the Transferase family), chromosome 9 (*evm.model.LG09.98* encoding cell division control protein), and chromosome 1 matching lipase. Interestingly, this SNP strongly influences the variation in IAA content but is not significantly associated with plant height (Figure 2b, Table 1).

### 2.3. WGCNA Unraveled the Molecular Basis of the Different Phenotypes in Diverse Bermudagrass

All the clean data of 91 bermudagrass samples across 16 latitudes were re-aligned with the African bermudagrass reference genome, and mapped genes were counted for further differential analysis. In order to better compare the performance of bermudagrass from different latitudes, the expression patterns in the L10 group (the intersection of two climatic zones) were assigned as the reference samples. A total of 3717 DEGs were identified from the union pool by the differential analysis of diverse bermudagrass compared to the L10 group. By GO enrichment analysis, these DEGs are mainly enriched in organic hydroxy compound metabolite (GO:1901615) of the biological process and coenzyme binding (GO:0050662) of the molecular process (Figure 3, Appendix A).

To understand the relationship between the function of DEGs and plant traits, WGCNA was used to analyze the association between the physiological data (plant height and IAA content) and 3717 DEGs identified from all comparisons. A total of 17 modules related to physiological indicators were observed (Figure 4a). Analysis of the module–trait relationships revealed that the ‘pink’ module of 133 genes was mostly positively correlated with HT (*r*^2^ = 0.37, *p* = 4 × 10^−4^), whereas it was negatively correlated with IAA (*r*^2^ = −0.25, *p* = 0.02). Therefore, we assumed these genes in the ‘pink’ module were affiliated with plant growth and the development of diverse bermudagrass.

To gain insight into the overall expression level of DEGs from the associated module, we further performed a differences analysis and compared samples from different latitudes to the control (Figure 4b). The results showed that many DEGs present higher expression levels in samples of mid-latitudes (L6-L11). These genes are involved in the thyroid hormone signaling pathway and terpenoid backbone biosynthesis, both implicated in regulating plant growth and development, such as genes encoding beta-glucosidase (BGLU31), glutathione peroxidase (GPX), mitogen-activated protein kinase 1/3 (MAPK1_3), and farnesol dehydrogenase (FLDH). However, a few genes, such as UDP-glucosyltransferase 73C (UGT73C), involved in zeatin biosynthesis, were down-regulated in samples of mid-latitude. These results indicate that many genes involved in multiple biological processes affect the plant height and IAA biosynthesis in diverse bermudagrass.

### 2.4. Analysis of Candidate Gene Associated with Plant Height and IAA Content in Diverse Bermudagrass

Based on the overlaid GWAS and WGCNA results, the *BGLU31* gene exhibited the SNP signal and differential expression in bermudagrass across a latitude gradient (Figure 4b, Table 1). Detailed in the gene expression of *CdBGLU31*, the fold change determined by qRT-PCR is fit closely by RNAseq data over most of the samples. Additionally, the correlation coefficient analysis showed that the relationship between *CdBGLU31* gene expression and plant height or IAA content could be divided according to climate zone (Figure 5 and Appendix A). Compared with the L10 samples, the *CdBGLU31* gene was down-regulated in the L1~L9 sample in the subtropical zone (Figure 5a). Interestingly, the fold change of *CdBGLU31* gene expression is negatively correlated with the magnitude of IAA content (*r*^2^ = −0.75, *p* = 0.02, Figure 5c), whereas there is no significant correlation with the magnitude of plant height (*r*^2^ = 0.54, *p* = 0.13, Figure 5b). Given that non-significant *p*-values (i.e., *p* ≥ 0.05) indicate an association only because of pleiotropy, both candidates affected plant height due to pleiotropy rather than linkage.

### 2.5. CdBGLU31 Is Likely a Key Gene for Plant Architecture and Growth Regulation in Bermudagrass

Firstly, we performed a phylogenetic analysis to understand the evolutionary relationship of *BGLU* genes. The evolutionary diversification between *CdBGLU31* and other *BGLU* from various plant species was deduced using phylogenetic analysis (Figure 6a). Notably, *CdBGLU31* in this study is closely clustered with the *BGLU31/32* gene of related species, including *Oryza sativa* and *Zea mays*. Additionally, the alignment of *CdBGLU31* was found with superfamily member PLN02849 (Glycosyl hydrolases) and is similar to the *BGLU11* gene of *Arabidopsis thaliana* (Appendix A).

Based on the above results, it is found that the different expressions of *CdBGLU3*1 in bermudagrass with diverse architecture and the correlation with IAA content are more prominent. We then sought to provide initial insight into the underlying molecular function of *CdBGLU31*. The phenotypic characterizations of the *CdBGLU31* homolog (*AtBGLU11*) mutant in Arabidopsis were analyzed. Two lines of T-DNA insertion mutants of Arabidopsis *bglu11* were obtained from stock centers of T-DNA insertion lines. The insertion mutation of *AtBGLU11* exon 8 produced lower height than those in the wild type (WT) in two investigated lines (Figure 6b,c). The rosette diameter of the mutant was also smaller than the WT, and the flowering was delayed (Appendix A). These results showed that *CdBGLU31* could be involved in regulating plant growth and development in bermudagrass.

## 3. Discussion

The plant architecture and growth habits of bermudagrass are strongly associated with application scenarios, and they can largely determine the practical value in the landscape, livestock, and eco-remediation [20,21,22]. Our previous studies have investigated the morphological diversity of a vast collection of bermudagrass. The collection is enriched by China, particularly subtropical and mesothermal zones, representing a wealth of local genetic resources that might be exploited in future conservation and breeding studies [3,23]. However, it was challenging and limited to study the regulatory mechanisms of bermudagrass plant type regulation without the genome information before the disclosure of the African bermudagrass genome sequence in August 2021 [7]. In this study, we dissected the genetic basis of plant type in bermudagrass through an integrative strategy by combining the GWAS and WGCNA approaches based on the RNAseq data.

Although RNAseq data are easily acquired in this “multi-omics” era, the whole genome sequence of non-model species with complex genetic backgrounds is difficult to obtain. Unlike many GWA studies that rely on genomic variation, the present study was based on the transcriptome variation and the genome of African bermudagrass. The RNAseq-based SNPs have been proven to be valuable for QTL mapping and functional and evolutionary studies [9,24,25]. Concurrently, our previous work on the diversity levels of phenotypic traits and variation at the RNAseq level was sufficient to detect numerous associations for GWAS [3,26]. Accordingly, the present study aimed to link the diversity observed at the phenotypic level at the transcribed genome level. We used three different GWAS approaches that provided multiple significant associations related to plant height and IAA content to achieve this goal. Among the methods used to perform these GWAS, FarmCPU and GLM shared high accuracy and reliability of associations, while the standard MLM method showed the least number of significant SNP signals (Figure 2). These data indicate that weaker associations might best be identified using multiple strategies.

The GWAS have mainly targeted genomic regions that underlie the genetic basis of complex traits in the grass family. Indeed, multiple genes of associations were detected with a valuable characteristic of grass with genome information, which was also successfully used in other studies [27,28,29]. In the present study, using GWAS on SNPs from RNAseq extracted from diverse bermudagrass, we did not expect a direct relationship for most of the present polymorphisms in the expressions of genes involved in plant architecture. However, we would suggest some of the candidate genes might either have pleiotropic effects on plant growth and development. To further decipher the regulatory mechanisms of candidate SNPs and genes involved in the plant type and IAA biosynthesis, we conducted WGCNA of transcriptomic data generated from 91 bermudagrass individuals. So, a beta-glucosidase gene from the glycosyl hydrolase family was both detected to have G→A SNP mutation from GWAS and expressed in different bermudagrass with diverse phenotypes by WGCNA (Table 1, Figure 4b).

Beta-glucosidase 31 (BGLU31) is a glycosyl hydrolase family GH1 that catalyzes glycosylation, controlling the reactivities and bioactivities of plant secondary metabolites and phytohormones, thereby regulating plant growth and development [30]. The numerous genes and orthologs encoding apparently functional GH1 protein have been identified in Arabidopsis and rice genome sequences [31]. In the current work, the *CdBGLU31* gene found in GWAS and WGCNA were clustered in the *OsBGLU31/32/33* group (Figure 6a). Luang et al. reported that *Os9BGlu31* is a vacuolar TG that could glycosylate the auxins indole acetic acid and naphthalene acetic acid [32]. It has recently been shown that the root length of the rice *bglu33* mutant responded to a 0.05 µM IAA and *Os9BGlu33* was up-regulated by exogenous IAA. The above studies suggest that *Os9BGlu31/33* plays specific and similar roles in root elongation with various relations with IAA signaling [33]. Together with our result of correlation coefficient analysis of *CdBGLU31* gene expression and IAA content (Figure 5c), it is reasonable to believe that the *CdBGLU31* gene could be involved in the IAA biosynthesis and signal transduction. In addition, IAA promotes root branching and interacts with ABA signals to regulate the development of the lateral root [34,35].

Interestingly, we found that the IAA content of L10 bermudagrass resources with shorter natural height and lateral growth type is the highest (Figure 2a). Moreover, the *CdBGLU31* gene was detected from the correlation between SNP and plant height, but there was a significant negative correlation between its expression and the magnitude of IAA content (Table 1, Figure 5c). In addition, analysis of Arabidopsis *bglu11* mutant also presents a lower plant height than the wild type. Although the *CdBGLU31* and *AtBGLU11* genes did not appear in the same clade in the phylogenetic tree, *CdBGLU31* shares 65% sequence identity with *AtBGLU11* (Figure 6a, Appendix A). The *AtBGLU11* gene was found in developing roots, leaves, and rosette plants [36]. Unfortunately, the functional analysis of *AtBGLU11* was not reported. Although these results could not conduct the direct relationship between the *CdBGLU31* gene or *Atbglu11* mutant and plant type, they are sufficient to predict that *CdBGLU31* is closely related to the regulation of plant type and IAA content.

Collectively, GWAS based on RNAseq data has been proven reliable in discovering SNPs at transcribed regions associated with quantitative traits. Hence, under the determination of the morphological diversity of 91 bermudagrass resources covering subtropical and mesothermal zones, an integrated approach linking GWAS with WGCNA was applied for the detection of candidate genes on the plant type and IAA biosynthesis. Then, one candidate gene (*CdBGLU31*) was found to have expression closely related to IAA content and had various relationships with the growth of plant type. The *CdBGLU31* gene probably regulated function of plant type, which might be useful for designing future gene-function, genomic-based breeding, and gene-editing studies of dwarf bermudagrass. Additionally, the unique accessions (L10) identified in this study could be used as pre-breeding materials to develop turf-type grass.

## 4. Materials and Methods

### 4.1. Plant Materials and Phenotypes

The bermudagrass (*Cynodon* spp.) materials were collected from 16 different wild sites in different regions of China. Six replicated tetraploids with similar growth status were selected from each latitude according to chromosomal observations to eliminate cytotype effects on SNPs and gene expression in this study. The final 91 individuals from different latitudes were used in the present study (Appendix A). All bermudagrass individuals were grown in a greenhouse on the campus of Yangzhou University. For individuals, three plants per replicate in a pot (20 cm depth × 16 cm diameter) were evaluated at the growth stage. Natural height (HT) was directly measured, and the content of IAA in the samples was determined using the fresh leaves with Plant Indole 3 Acetic Acid (IAA) ELISA Kit (mlBio, Shanghai, China) according to the manufacturer’s protocol.

### 4.2. RNA Sequencing, SNP Calling, and Gene Expression

RNA samples from each individual were isolated using TRIzol according to the manufacturer’s protocol. RNA extracts were used to create cDNA libraries, and cDNA sequencing was performed using an Illumina NovaSeq. Following the removal of adaptor sequences and low-quality reads, raw data (PRJNA646313) were assembled into the genome of African bermudagrass (PRJCA003581 in China National Center for Bioinformation GSA) with hisat2 alignment [37]. SAMtools (v 1.12) were used to switch the sam to bam files [38]. By using Picard (v 2.9.0), the bam files were sorted, and duplication reads were removed. With GATK (version 4.1.9.0), ‘N’ trims are split, variants are filtered, and SNPs are called [39]. Around 1,636,265,647 raw SNPs were identified in the 91 bermudagrass individuals. By filtering and keeping the SNPs with minor allele frequency (MAF) > 2% and missing allele <30%, 4,751,646 SNPs were retained and distributed on nine chromosomes (Chr) that were used in this study (Appendix A). Concurrently, the mapped fragments after assembly for each gene were counted by featureCounts. Genes with average read counts were considered expressed, and gene ontology (GO) analysis was conducted.

### 4.3. Association Analysis

*GWAS*: All 4,751,646 SNPs and plant traits (HT and IAA content) were used to perform GWAS by a memory-efficient, visualization-enhanced, and parallel-accelerated tool (rMVP) with a general linear model (GLM), a mixed linear model (MLM), and a fixed and random model circulating probability unification (FarmCPU mode) in R software [32,40,41,42]. The significant threshold of associations was calculated using a Bonferroni correction of *p*-value with an α = 0.05 (0.05/SNP number). The results were shown on the -log scale in Manhattan. We highlighted SNPs with a *p*-value < 0.05 in the Manhattan plot to indicate other regions that could potentially be associated with the traits. In order to predict its functional impact, we need to place the SNP in the genomic context via IGV software (v 2.11.3).

*WGCNA*: After discarding undetectable or relatively low expression genes, as the sample from the L10 site with the control, differentially expressed genes (DEGs) were used for the WGCNA package in R. The default WGCNA ‘step-by-step network construction’ analysis was used to build the modules. We calculated the adjacency relationship between genes and constructed a topological matrix. A hierarchical clustering tree with the dissimilarity of the topological matrix was generated, and the modules of dynamic trees were cut by a standard method. Then, similar modules were merged by calculating the eigengenes module. Finally, a cluster dendrogram was formed with a soft power of 6, a minimum module size of 50 genes, and a distance threshold cut of 0.1 (Appendix A).

### 4.4. Functional Analysis of Candidate Genes

Verification of qRT-PCR: Total RNAs were extracted as described above for the samples used for RNA-seq. The qRT-PCR analysis was performed to validate the reliability of the RNA-seq data. The candidate gene of *beta-glucosidase 31* was chosen for the qRT-PCR study with the forward primer (AGCGTTTGGGACACCTTC) and reverse primer (AGCCCTTTGGATTGATTTCT). Each sample at different latitudes had three replicates. The qRT-PCR assays were performed using the SYBR Mix Kit (TransGen Biotech, Beijing, China) and were conducted on a Roche LightCycler 96 Sequence Detection System, with a reaction for 10 min at 95 °C followed by 40 amplification cycles of 10 s at 95 °C, 30 s at 55 °C, and 30 s at 72 °C. The reference gene, *CdActin*, was used to normalize the expression levels of target candidate genes. The expression levels were calculated using the 2^−ΔΔCt^ method.

Correlation analysis: The relationship between candidate gene expression from qRT-PCR and plant height or IAA content in different samples was studied using a ggcorrplot package with the Pearson method in Rstudio software.

Arabidopsis mutant analysis: On the basis of obtaining candidate gene sequences, a phylogenetic tree was constructed with orthologous genes of *Arabidopsis thaliana*, *Oryza sativa*, *Zea mays*, and the candidate gene in this study. Then, Arabidopsis mutants (SAIL_62_D08) were purchased according to the results of phylogenetic analysis. After sowing and identifying homozygous mutants, we observed and measured the phenotypes.

## Figures and Tables

**Figure 1 ijms-23-11432-f001:**
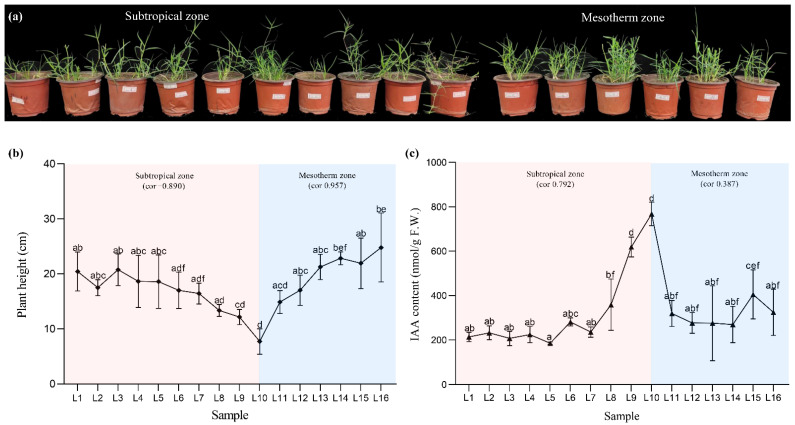
Phenotypic analysis of diverse bermudagrass across a latitudinal gradient. (**a**) Plant morphologies of wild germplasm accessions of bermudagrass. From right to left: L1~L16 (L represents location). (**b**) Natural plant height of tested samples of bermudagrass before mowing. (**c**) IAA content of diverse bermudagrass. Means of three replicates ± standard error is shown. The data were subjected to an ANOVA test to determine the LSD, and bars superscribed by different letters are significantly different at *p* < 0.05.

**Figure 2 ijms-23-11432-f002:**
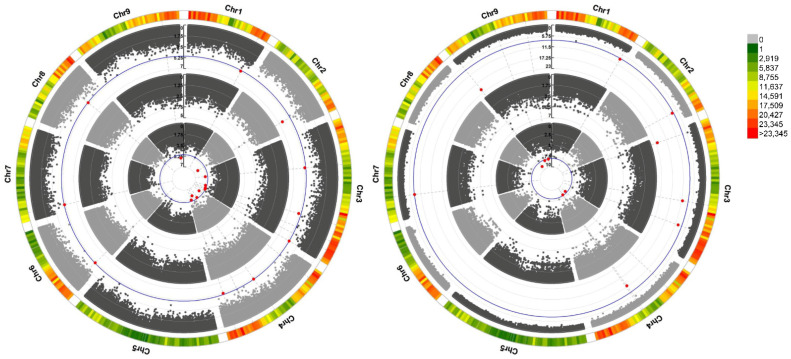
Genome-wide association study of phenotypic trait with SNPs. Manhattan plot shows association signals with (**a**) plant height and (**b**) IAA content. The blue lines show the threshold, and the red point represents the significant signals with the corresponding trait. The outermost circle indicates the SNP density.

**Figure 3 ijms-23-11432-f003:**
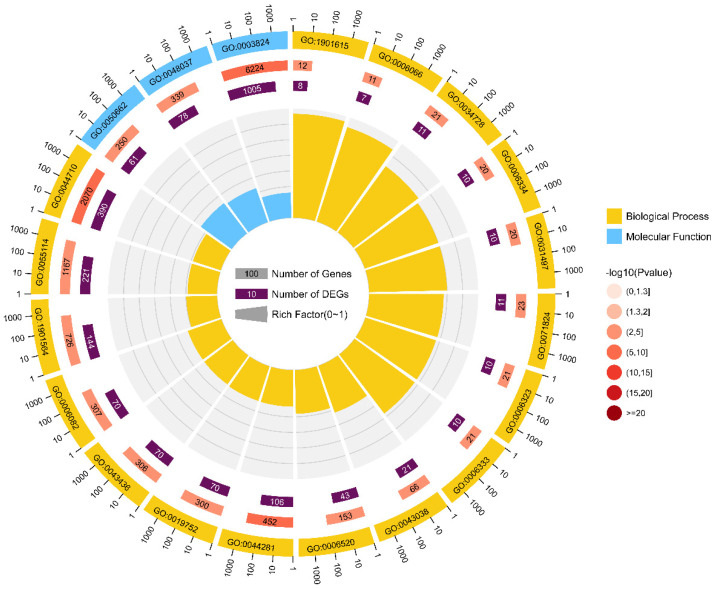
GO enrichment analysis of significant genes identified by RNAseq. The count number inside the orange-red and purple box represents the number of genes and DEGs enriched in a process, respectively, and the color represents the range of the *p*-value.

**Figure 4 ijms-23-11432-f004:**
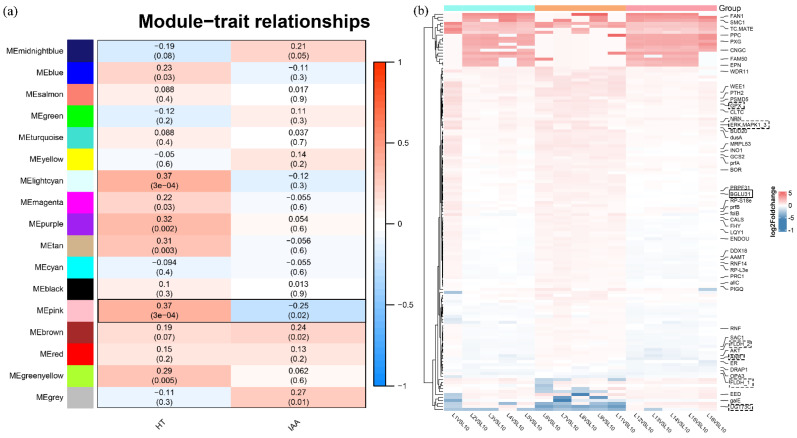
Expression analysis of DEGs in diverse bermudagrass across a latitude gradient. (**a**) Gene expression modules of WGCNA correlated with bermudagrass’s quantitative traits, including plant height (HT) and IAA content. We report the Pearson coefficient and its *p*-value. Highly positive correlations are shown in red, and highly negative correlations are shown in blue. The combination of module–trait marked in a black frame will be selected for subsequent expression and annotation analysis. (**b**) Transcript-level difference of module genes in bermudagrass from different latitudes measured by comparisons of Log2 (sample from different latitude counts/L10 sample counts).

**Figure 5 ijms-23-11432-f005:**
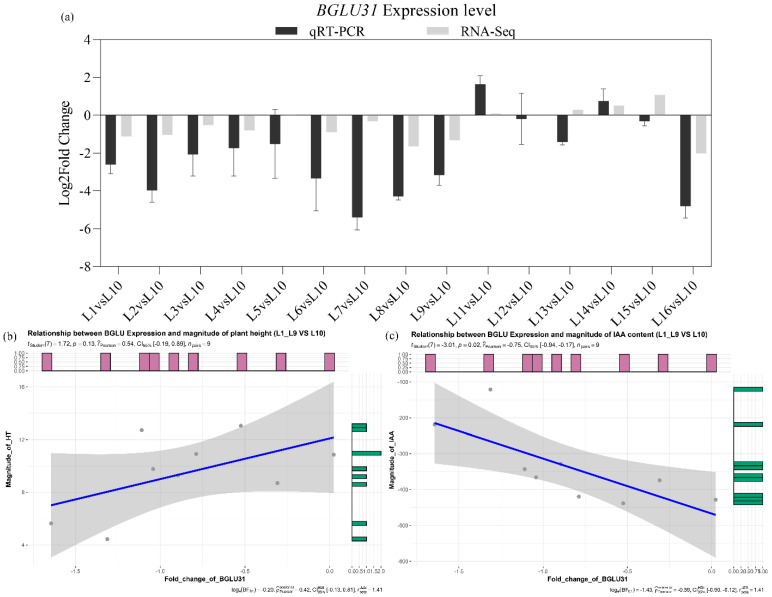
Gene expression and correlation coefficient analysis of *CdBGLU31*. (**a**) Expression validation of *CdBGLU31* between qRT-PCR and RNA-seq analysis. The L10 sample is the control reference, and the relative expression was determined by log2 FPKM and Log2 (fold change) compared to the control. All qRT-PCR reactions were performed from triplicate biological samples. The mean of three replicates ± standard error is shown. (**b**) Relationship between *CdBGLU31* expression and magnitude of plant height (L1~L9 vs. L10). (**c**) Relationship between *CdBGLU31* expression and magnitude of IAA content (L1~L9 vs. L10).

**Figure 6 ijms-23-11432-f006:**
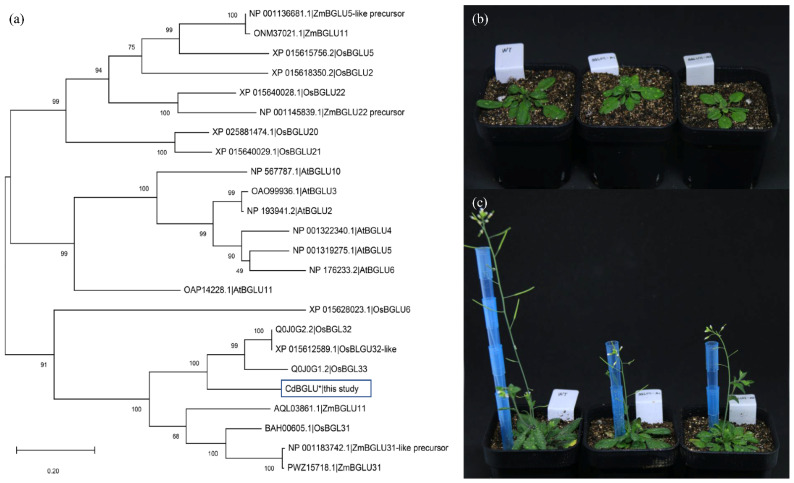
Functional analysis of CdBGLU31. (**a**) Analysis of phylogenetic relationships among *BGLU* gene. Os, *Oryza sativa*; Zm, *Zea mays*; At, *Arabidopsis thaliana*. The black box indicates the bermudagrass gene isolated in this study. (**b**,**c**) The morphology of Arabidopsis wild type and bglu11 mutants in vegetative growth and heading stage.

**Table 1 ijms-23-11432-t001:** SNPs were significantly associated with plant height and IAA content, and a subset of candidate genes was identified.

SNP	Chr.	Position	*p*	Allele	Candidate Gene ID	Pfam_Annotation
**Associate with Plant Height**						
LG01-30416755	1	30416755	7 × 10^−6^	C/T	evm.model.LG01.2990	Unknown
LG03-18657355	3	18657355	4.2 × 10^−6^	T/C	evm.model.LG03.1638	CytoLGome b5-like Heme/Steroid
LG03-44157320	3	44157320	6.1 × 10^−6^	A/C	evm.model.LG03.3214	TMEM33/Pom33 family
LG03-44157337	3	44157337	6.1 × 10^−6^	T/G	evm.model.LG03.3214	TMEM33/Pom33 family
LG03-51374819	3	51374819	8.2 × 10^−6^	G/A	evm.model.LG03.4036	Protein of unknown function (DUF620)
LG04-34820631	4	34820631	1.6 × 10^−6^	A/G	evm.model.LG04.2874	GTP cyclohydrolase II
LG04-4787029	4	4787029	1.8 × 10^−6^	A/G	evm.model.LG04.571	Surface antigen
LG04-42658778	4	42658778	6.1 × 10^−6^	A/G	evm.model.LG04.3796	TLD
LG04-24215127	4	24215127	6.9 × 10^−6^	G/A	evm.model.LG04.1979	Serine-threonine/tyrosine-protein kinase
LG06-3448481	6	3448481	8.8 × 10^−6^	T/C	evm.model.LG06.412	Ring finger domain98
LG09-36689561	9	36689561	3.0 × 10^−6^	G/A	evm.model.LG09.2890	Beta-glucosidase
**Associate with IAA Content**						
LG01-32609814	1	32609814	3.7 × 10^−9^	G/A	evm.model.LG01.3294	Lipase (class 3)
LG04-24619228	4	24619228	7.0 × 10^−9^	G/A	evm.model.LG04.1998	Calcineurin-like phosphoesterase
LG04-23975106	4	23975106	1.0 × 10^−12^	A/T	evm.model.LG04.1964	Transferase family
LG07-10265433	7	10265433	7.5 × 10^−9^	C/T	evm.model.LG07.712	Unknown
LG09-1324439	9	1324439	7.8 × 10^−9^	A/G	evm.model.LG09.98	Cell division control protein 14, SIN component

## Data Availability

The datasets presented in this study are available in Tables, Figures, Appendix A, and Appendix A. The accession number(s) used in this study can be found in the article or Appendix A.

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
