# Peer review of "A Novel Beta-Glucosidase Gene for Plant Type Was Identified by Genome-Wide Association Study and Gene Co-Expression Analysis in Widespread Bermudagrass"

_ijms, 2022, doi:10.3390/ijms231911432_

Round 1
Reviewer 1 Report
This work presents a potential gene candidate for differentiating bermudagrass populations/varieties. Overall the manuscript reads well; however, all figures require some work - in some figures, the text is not visible, and in others, labels of the x-axis and y-axis contain underscores. Please look for such errors in all figures. The title should be shortened.
Author Response
Dear Editor and Reviewers:
Thanks for your letter and the reviewers’ comments concerning our manuscript entitled “A novel candidate beta-glucosidase gene for plant type was identified by genome-wide association study and gene co-expression analysis in natural population of bermudagrass”(ID: ijms-1906163). Those comments are all valuable and very helpful for revising and improving our paper, as well as the important guiding significance to our researches. We have studied the comments carefully and would like to revise them for your consideration. We have addressed the comments raised by the reviewers, and the amendments are highlighted yellow or track change mode in the revised manuscript.
Reviewer #1:
General comments:
This work presents a potential gene candidate for differentiating bermudagrass populations/varieties. Overall the manuscript reads well; however, all figures require some work - in some figures, the text is not visible, and in others, labels of the x-axis and y-axis contain underscores. Please look for such errors in all figures. The title should be shortened.
Answer: Thanks for the positive comments.
We are sorry that there could be misunderstanding and errors in all figures, probably because of the conversion after the figure was inserted into the main text. Furthermore, I have checked and revised the formats of all figures, and submit the original high-quality pictures separately by email to ensure the quality and correct display.
For the title, we have revised it to “A novel beta-glucosidase gene for plant type was identified by genome-wide association study and gene co-expression analysis in wildspread bermudagrass”
We hope the revised manuscript could be acceptable for the journal. Also, we look forward to hearing from you regarding our revision. We would be glad to respond to any further questions and comments that you may have.
Sincerely,
Lu Gan
Reviewer 2 Report
The paper presented for review is very interesting and concerns the identification of a new beta-glucosidase gene, potentially related to the growth and biosynthesis of IAA in bermudagrass. The manuscript is written in intresting for readers way and the obtained results seem to be innovative. I have a few more specific comments:
1) why are the words shoots and stolons repeated in parentheses on lines 58 and 59?
2) the primer sequence and qRT-PCR conditions were not provided in the section: Functional analysis of canditate genes
3) in the graph (Fig. 1c) the IAA content should be expressed in nmol/g, but fresh or dry weight? I also suggest specifying in the Materials and methods the amount of tissue used for IAA determination
4) the quality of the figures could be better, especially 2, 3 and 4, the inscriptions are illegible
5) a list of literature should be prepared in accordance with the guidelines of the journal
Author Response
Dear Editor and Reviewers:
Thanks for your letter and the reviewers’ comments concerning our manuscript entitled “A novel candidate beta-glucosidase gene for plant type was identified by genome-wide association study and gene co-expression analysis in natural population of bermudagrass” (ID: ijms-1906163). Those comments are all valuable and very helpful for revising and improving our paper, as well as the important guiding significance to our researchers. We have studied the comments carefully and would like to revise them for your consideration. We have addressed the comments raised by the reviewers, and the amendments are highlighted yellow or track change mode in the revised manuscript.
General comments:
The paper presented for review is very interesting and concerns the identification of a new beta-glucosidase gene, potentially related to the growth and biosynthesis of IAA in bermudagrass. The manuscript is written in interesting for readers way and the obtained results seem to be innovative.
Answer: Thanks for your positive comments.
Specific comments:
- Why are the words shoots and stolons repeated in parentheses on lines 58 and 59?
Answer: Thank you for you correction. I am sorry for the error and have changed it. In order to avoid the recurrence of such errors, we have comprehensively polished and revised the manu. The modifications are highlighted yellow or track change mode in the revised manuscript.
- the primer sequence and qRT-PCR conditions were not provided in the section: Functional analysis of candidate genes.
Answer: Good point. We have added it (Ln127-132).
- in the graph (Fig. 1c) the IAA content should be expressed in nmol/g, but fresh or dry weight? I also suggest specifying in the Materials and methods the amount of tissue used for IAA determination
Answer: Okay. We have modified it (Ln 84 and Figure 1c).
- the quality of the figures could be better, especially 2, 3 and 4, the inscriptions are illegible.
Answer: Indeed, the figures inserted into word is compressed by converted pixels, and the quality is reduced. I will send all original high-quality figures to the editorial office by email.
- a list of literature should be prepared in accordance with the guidelines of the journal.
Answer: Okay. We have carefully modified references format and all the main text using the Microsoft word template.
We hope the revised manuscript could be acceptable for the journal. Also, we look forward to hearing from you regarding our revision. We would be glad to respond to any further questions and comments that you may have.
Sincerely,
Lu Gan